# Collagen IV and basement membrane at the evolutionary dawn of metazoan tissues

**Aaron L Fidler**[1,2,3†], **Carl E Darris**[1†], **Sergei V Chetyrkin**[1,4†], **Vadim K Pedchenko**[1,4], **Sergei P Boudko**[1,4], **Kyle L Brown**[1,4,5], **W Gray Jerome**[6], **Julie K Hudson**[2,7], **Antonis Rokas**[8], **Billy G Hudson**[1,2,4,6,9,10,11,12]*

[1]Department of Medicine, Division of Nephrology and Hypertension, Vanderbilt University Medical Center, Nashville, United States; [2]Aspirnaut Program, Vanderbilt University Medical Center, Nashville, United States; [3]Department of Biological Sciences, Tennessee State University, Nashville, United States; [4]Center for Matrix Biology, Vanderbilt University Medical Center, Nashville, United States; [5]Center for Structural Biology, Vanderbilt University Medical Center, Nashville, United States; [6]Department of Pathology, Microbiology, and Immunology, Vanderbilt University Medical Center, Nashville, United States; [7]Department of Medical Education and Administration, Vanderbilt University Medical Center, Nashville, United States; [8]Department of Biological Sciences, Vanderbilt University Medical Center, Nashville, United States; [9]Department of Cell and Developmental Biology, Vanderbilt University Medical Center, Nashville, United States; [10]Department of Biochemistry, Vanderbilt University Medical Center, Nashville, United States; [11]Vanderbilt-Ingram Cancer Center, Vanderbilt University Medical Center, Nashville, United States; [12]Vanderbilt Institute of Chemical Biology, Vanderbilt University Medical Center, Nashville, United States

*For correspondence: billy.
hudson@vanderbilt.edu

†These authors contributed
equally to this work

Competing interest: See
page 21

Reviewing editor: Harry C
Dietz, Howard Hughes Medical
Institute and Institute of Genetic
Medicine, Johns Hopkins
University School of Medicine,
United States

**Abstract** The role of the cellular microenvironment in enabling metazoan tissue genesis remains obscure. Ctenophora has recently emerged as one of the earliest-branching extant animal phyla, providing a unique opportunity to explore the evolutionary role of the cellular microenvironment in tissue genesis. Here, we characterized the extracellular matrix (ECM), with a focus on collagen IV and its variant, spongin short-chain collagens, of non-bilaterian animal phyla. We identified basement membrane (BM) and collagen IV in Ctenophora, and show that the structural and genomic features of collagen IV are homologous to those of non-bilaterian animal phyla and Bilateria. Yet, ctenophore features are more diverse and distinct, expressing up to twenty genes compared to six in vertebrates. Moreover, collagen IV is absent in unicellular sister-groups. Collectively, we conclude that collagen IV and its variant, spongin, are primordial components of the extracellular microenvironment, and as a component of BM, collagen IV enabled the assembly of a fundamental architectural unit for multicellular tissue genesis.

## Introduction

A pivotal event in metazoan evolution was the transition from single-cell organisms to multicellular tissues (**Figure 1A**). The cellular microenvironment is presumed to play an essential role in this transition, yet the mechanism remains obscure. The basement membrane (BM), a specialized form of extracellular matrix (ECM), is a hallmark morphological feature of the microenvironment of epithelial

**eLife digest** The emergence of the diversity of multicellular animals involved cells joining together to form tissues and organs. The 'glue' that enabled the cells to work together is made of rope-like molecules called collagen, which assemble into scaffolds. These smart scaffolds tether proteins forming basement membranes that connect cells, provide strength to tissues, and transmit information that influences how the cells behave.

How did collagen evolve over millions of years to enable the ever-increasing complexity, size and diversity of animals? To investigate, Fidler, Darris, Chetyrkin et al. explored the tissues of the most ancient of currently living animals – the comb jellies and sponges. This revealed that among all the collagens that make up the human body, a type called collagen IV was a key innovation that enabled single celled organisms to evolve into multicellular animals. Collagen IV, as molecular glue, enabled the formation of a fundamental architectural unit of basement membrane and cells that allowed multicellular tissues and organs to evolve.

The findings presented by Fidler, Darris, Chetyrkin et al. pose questions about how collagen IV glues cells together, and how information is stored in the rope-like scaffolds to influence cell behavior. Understanding these processes could ultimately lead to the development of new treatments for diseases in which the collagen smart scaffolds play a key role, such as in kidney diseases and cancer.

tissues, and its appearance within the non-bilaterian animal phyla suggests it was a prerequisite (*Sherwood, 2015*; *Hynes, 2012*; *Ozbek et al., 2010*). The BM has numerous functions including maintaining tissue architecture and compartmentalization, organizing growth factor signaling gradients, guiding cell migration and adhesion, delineating apical-basal polarity modulating cell differentiation during development, orchestrating cell behavior in tissue repair after injury, and guiding organ regeneration (*Hynes, 2009*; *Yurchenco, 2011*; *Vracko, 1974*; *Pöschl et al., 2004*; *Daley and Yamada, 2013*; *Wang et al., 2008*; *Pastor-Pareja and Xu, 2011*; *Song and Ott, 2011*).

The basement membrane is a supramolecular scaffold, comprised of a toolkit of proteins including collagen IV, laminin, perlecan, and nidogen (*Hynes, 2012*; *Fahey and Degnan, 2010*). Among these proteins, recent studies reveal collagen IV is an ancient protein with up to six distinct genes (COL4A1, COL4A2, COL4A3, COL4A4, COL4A5, COL4A6), essential for early development, that functions as a smart scaffold providing tensile strength to tissues, influencing cell behavior by tethering diverse macromolecules, including laminin, proteoglycans, growth factors, binding integrins (*Gupta et al., 1997*; *Bhave et al., 2012*; *McCall et al., 2014*; *Fidler et al., 2014*; *Pöschl et al., 2004*; *Vanacore et al., 2009*; *Cummings et al., 2016*; *Wang et al., 2008*; *Parkin et al., 2011*; *Emsley et al., 2000*). Disrupting collagen IV scaffolds causes BM destabilization and tissue dysfunction in mice, zebrafish, flies, and nematodes (*Pöschl et al., 2004*; *Fidler et al., 2014*; *Borchiellini et al., 1996*; *Gupta et al., 1997*). Collectively, these findings reveal that collagen IV, a component of the cellular microenvironment, is essential for tissue architecture and function; yet, the origin and molecular evolution of collagen IV remains obscure.

Knowledge of collagen IV evolution may shed light on the fundamental features of the cellular microenvironment that enabled the transition from single-cell organisms to multicellular tissues. Together, the non-bilaterian animal phyla (Ctenophora, Porifera, Placozoa, and Cnidaria) represent this transition. Importantly, Ctenophora has recently emerged as one of the earliest-branching extant phyla (*Ryan et al., 2013*; *Moroz et al., 2014*; *Whelan et al., 2015*; *Telford et al., 2016*), along with the sponges (Porifera) (*Pisani et al., 2015*; *Jékely et al., 2015*; *Telford et al., 2016*). Here, we sought to identify ECM components in Ctenophora along with the other non-bilaterian animal phyla, and compared the components to Bilateria and the metazoan sister-groups, Choanozoa, Filasterea, Amoebozoa, and Apusozoa. Our findings reveal that collagen IV and its truncated variant, spongin, are associated with the transition to multicellularity, and further that collagen IV, as a component of BM scaffolds, enabled the genesis of multicellular epithelial tissues.

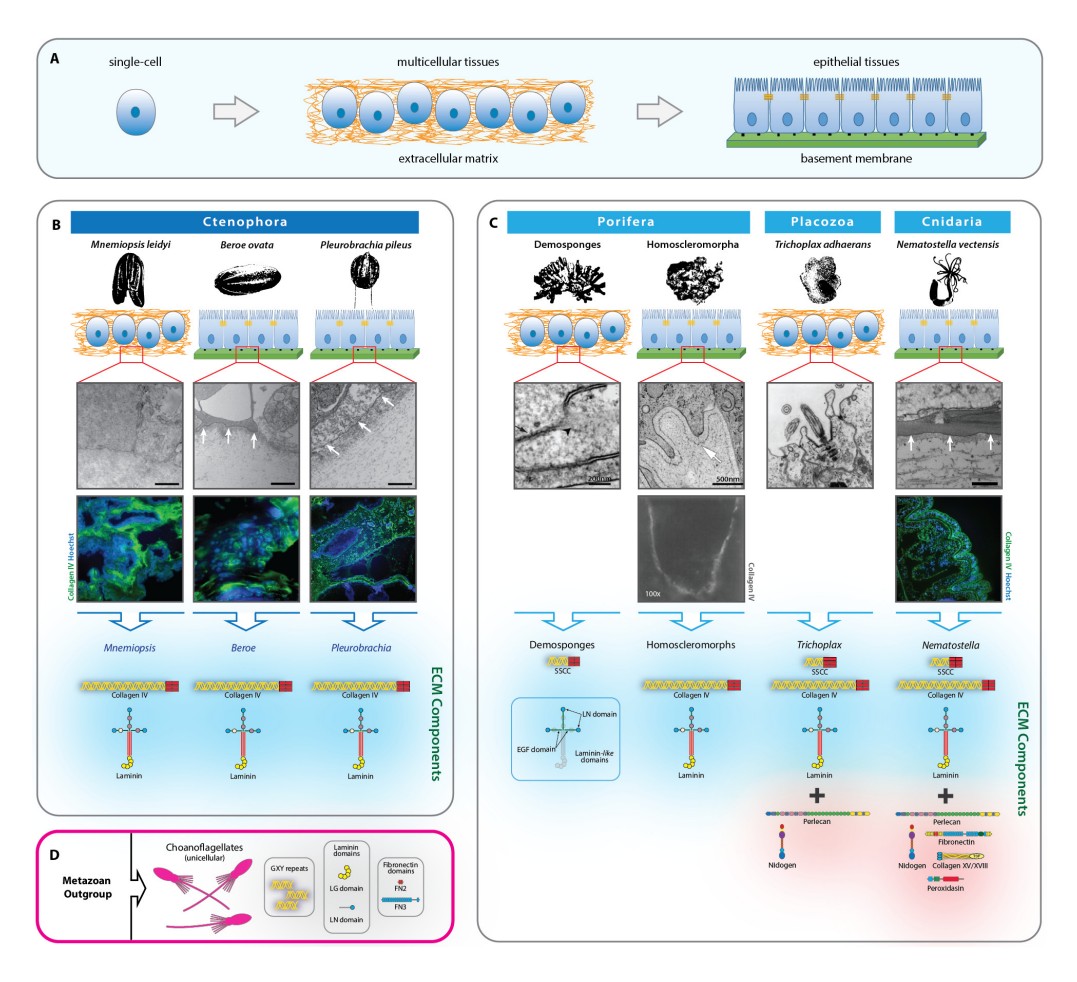

**Figure 1.** Extracellular matrix of the non-bilaterian animal phyla. (**A**) The transition from single-cell organisms to complex multicellular animals was enabled by an extracellular matrix. (**B**) Electron microscopy (EM) and immunohistochemistry (IHC) of the Ctenophora species, *Mnemiopsis* (IHC: 20X magnification), *Pleurobrachia* (IHC: 20X magnification), and *Beroe* (IHC: 40X magnification) and ECM components of Ctenophora. (**C**) Electron microscopy (EM) and immunohistochemistry (IHC) of the non-bilaterian animal phyla, Cnidaria (*Nematostella*; 20X magnification), Placozoa (*Trichoplax*), and Porifera (Homoscleromorpha and Demosponges) and ECM components of Porifera, Placozoa, and Cnidaria. Demosponge EM reproduced from *Figure 1E* of Adams, *et al.*, *Freshwater Sponges Have Functional, Sealing Epithelia with High Transepithelial Resistance and Negative Transepithelial Potential*, PLoS ONE, 2010, volume 5; Homoscleromorph EM reproduced from *Figure 3B*, Leys *et al.*, *Epithelia and integration in sponges*, Integrative and Comparative Biology, 2009, volume 49 with permission from Oxford University Press; Homoscleromorph IHC reproduced from Boute *et al.*, *Type IV collagen in sponges, the missing link in basement membrane ubiquity*, Biology of the Cell, 1996, volume 88 with permission from Wiley; Trichoplax EM reproduced from Ruthmann *et al.*, *The ventral epithelium of Trichoplax adhaerens (Placozoa): Cytoskeletal structures, cell contacts and endocytosis*, Zoomorphology, 1986, volume 106 with permission from Springer. (**D**) ECM components in choanoflagellates, the unicellular sister-group to metazoa. All scale bars 500 nm, unless otherwise noted.

## Results

### Ctenophore extracellular matrix is distinct from other metazoans

We characterized the extracellular matrix in Ctenophora (comb jellies) and the other non-bilaterian animal phyla through a combination of immunohistochemistry (IHC), electron microscopy (EM), RNA sequencing, and genomic and transcriptomic analyses. Three ctenophore species, *Mnemiopsis leidyi*, *Beroe ovata*, and *Pleurobrachia pileus*, were used for EM and IHC experiments. Systematic assessment by EM of a number of sections from similar areas in *Mnemiopsis* was conducted, and no organized basement membrane was encountered. Furthermore, tight junctions between cells and cellular polarization were not observed, both hallmarks of epithelial basement membrane tissue structure

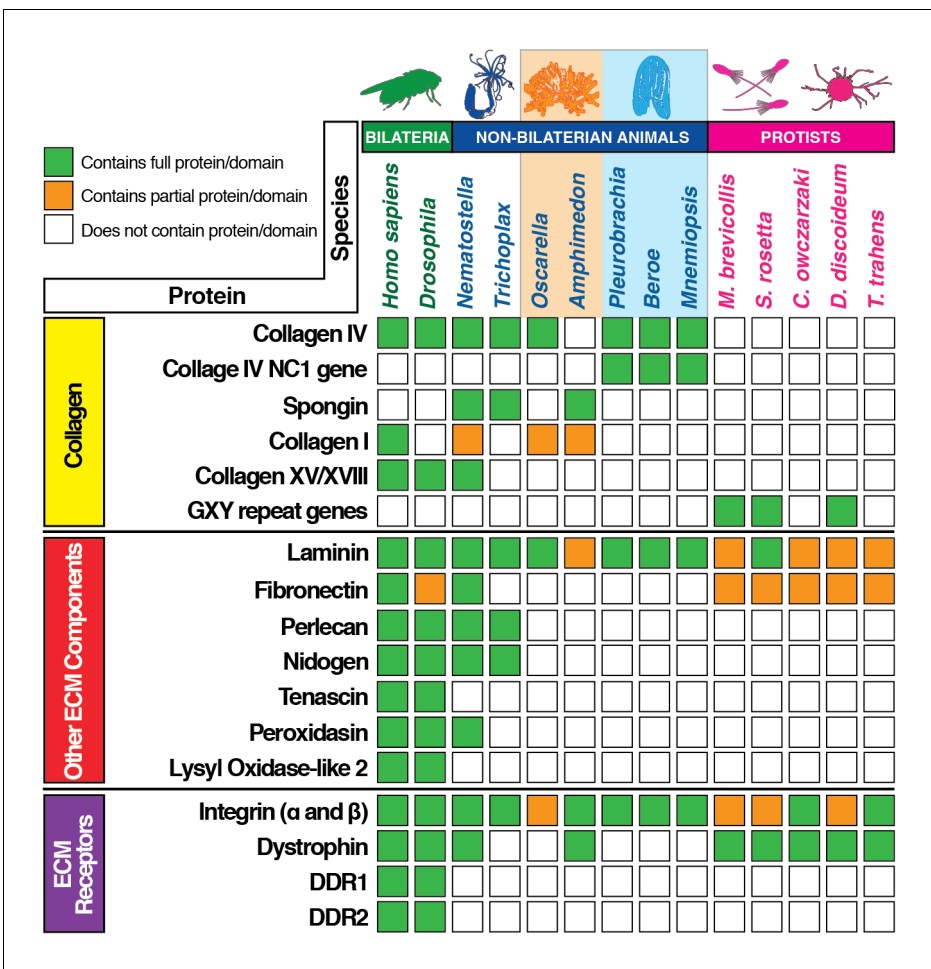

**Figure 2.** Extracellular matrix gene content across bilaterian, non-bilaterian animal, and unicellular protist phyla. Protein BLAST searches using the human ortholog of each protein as bait was conducted for ECM gene content analysis. Where possible (with exception of ctenophore species), we performed a search by protein name across each database. The databases used were Ensembl (http://protists.ensembl.org), NeuroBase (http://neurobase.rc. ufl.edu), AmoebaDB (http://amoebadb.org) and NCBI's Blast (https://blast.ncbi.nlm.nih.gov/Blast.cgi). Complete hits are denoted in green, while partial protein or domain sequences are denoted in orange. White boxes indicate absence of that protein/domain.

(*Figure 1B*). IHC was congruent with this finding, indicating that collagen IV was dispersed through-out the tissue, surrounding and encompassing cells. In *Beroe* and *Pleurobrachia*, however, EM indicated an electron dense layer underlying cells along with cell polarization, and lateral tight junctions between cells, and IHC similarly showed a dense collagen IV layer underlying cells (*Figure 1B*). Together, these features are congruent with basement membrane architecture and epithelial tissue. Basement membrane structures are prevalent throughout metazoa, from Cnidaria to vertebrates, and we sought to compare the basement membrane architecture in Ctenophora to that of other non-bilaterian animal phyla and Bilateria. *Nematostella*, along with other cnidarians, have a bilayer body structure composed of endoderm and ectoderm layer with an intervening mesoglea; however, the general BM structure is congruent with that of bilaterian organisms, including mammals. *Nematostella* demonstrated presence of basement membrane, characterized by polarized cells apical to an electron dense layer by EM, and a concentrated region of collagen IV underlying cell nuclei by IHC (*Figure 1C*).

We then characterized the ECM composition through analysis of transcriptomic and genomic data across the non-bilaterian animal phyla in comparison with Bilateria and unicellular sister-groups.

Ctenophore genomic and transcriptomic data were publicly available from the Pleurobrachia Genome Browser on Neurobase (http://neurobase.rc.ufl.edu/Pleurobrachia) and the Mnemiopsis Genome Project Portal on the National Human Genome Research Institute site (https://kona.nhgri. nih.gov/mnemiopsis/). The ECM components in *Nematostella* are very similar to that of most bilaterian species BMs (*Hynes, 2012*), including human, mouse, zebrafish, *Drosophila*, and *C. elegans*, consisting of collagen IV, laminin, peroxidasin, collagen XV and XVIII, perlecan, nidogen, fibronectin, as well as spongin (*Figure 1C* and *Figure 2*). Ctenophora, however, revealed a simplified set of ECM proteins, with collagen IV and laminin as the only components identified across *Beroe, Pleurobrachia,* and *Mnemiopsis* (*Figure 1B* and *Figure 2*). Importantly, despite lacking the full gamut of bilaterian ECM proteins, ctenophore cells can still construct a prototypical basement membrane in *Pleurobrachia* and *Beroe. Mnemiopsis*, however, does not form a BM despite exhibiting the very same toolkit proteins and this may be a result of secondary loss event.

Across Porifera and Placozoa, basement membranes are uncommon. Placozoa and the poriferan classes of calcareous and demosponges lack basement membranes (*Figure 1C*) (*Ozbek et al., 2010*; *Leys et al., 2009*; *Ruthmann et al., 1986*; *Srivastava et al., 2010*), suggesting the BM may have been secondarily lost in these lineages (*Cock, 2010*) or that it is present only at specific stages during their life cycle (*Hynes, 2012*). Alternatively, basement membranes may have independently evolved in Ctenophora, Porifera, and Bilateria, a phenomenon that could have occurred because of shared inheritance of ECM proteins and domains from the last common ancestor of the non-bilaterian animal phyla. Investigation of the non-bilaterian animal phyla for components of the BM toolkit suggests that many of the components are present (*Srivastava et al., 2010*, *2008*). Specifically, Placozoa contains the necessary components for a BM, including collagen IV, laminin, perlecan, and nidogen (*Srivastava et al., 2008*) and also exhibits spongin (*Figure 1C* and *Figure 2*) (*vide infra*). In Porifera, the ECM of homoscleromorph sponges contains basement membranes (*Boute et al., 1996*), with collagen IV and laminin, but no spongin (*Figure 1C* and *Figure 2*). Demosponges, on the other hand, lack any detectable collagen IV and only contain laminin-related domains and spongin. In contrast, the Demosponge and Hexactinellidae classes of Porifera, which both lack collagen IV, do not have basement membranes (*Figure 1C*) (*Adams et al., 2010*).

Laminin architecture appears to be present in choanoflagellates (*Fahey and Degnan, 2012*), and additionally laminin-related genes appear in all other unicellular species including *Capsaspora owczarzaki*, *Dictyostelium discoideum*, and *Thecamonas trahens* (*Figure 2*). Although no other complete ECM components exist in unicellular choanoflagellates (*King, 2005*), several domains and fragments of ECM components have been identified, including collagenous repeats, laminin G (globular) domains and LN (N-terminal) domains, and fibronectin type II and III domains (*Figure 1D*) (*King et al., 2008*). Furthermore, no other complete ECM components exist in other unicellular organisms, including *Salpingoeca rosetta* (Choanozoa), *Capsaspora owczarzaki* (Filasterea), *Dictyostelium discoideum* (Amoebozoa), or *Thecamonas trahens* (Apusozoa); however, collagenous GXY repeats were identified in *Monosiga, Salpingoeca,* and *Dictyostelium*. Receptors for collagen binding, including integrin, dystrophin, and Discoidin domain receptors (DDRs) were analyzed across metazoa and unicellular protists (*Figure 2*). Integrins are highly conserved across metazoa and are found in the unicellular *Capsaspora* and *Thecamonas*. Dystrophin was identified in humans, fruit fly, *Nematostella, Amphimedon*, as well as in unicellular protists but is absent in *Trichoplax, Oscarella*, and ctenophores. DDR 1 and 2 was identified only in humans and fruit fly. Collectively, the ECM composition across the non-bilaterian animal phyla points to laminin and collagen IV as highly conserved components, and importantly, that they are associated with BM and epithelial tissue architecture.

## Ctenophora is distinct from all other metazoans in number and organization of collagen IV genes

We characterized the gene and protein structure of collagen IV in Ctenophora and the other non-bilaterian animal phyla. In *Mnemiopsis*, 11 full-length collagen IV genes were discovered, which contrasts with the two genes typically present throughout invertebrates and the six genes typically found in vertebrates (*Khoshnoodi et al., 2008*). The head-to-head orientation is a distinguishing characteristic of collagen IV genes among other collagens of Bilateria (*Kaytes et al., 1988*; *Hudson et al., 1993*). In *Mnemiopsis*, four genes occur on the same scaffold and exhibit a head-to-head orientation (ML17501a and ML17504a, ML17502a and ML17503a) (designated as Group I)

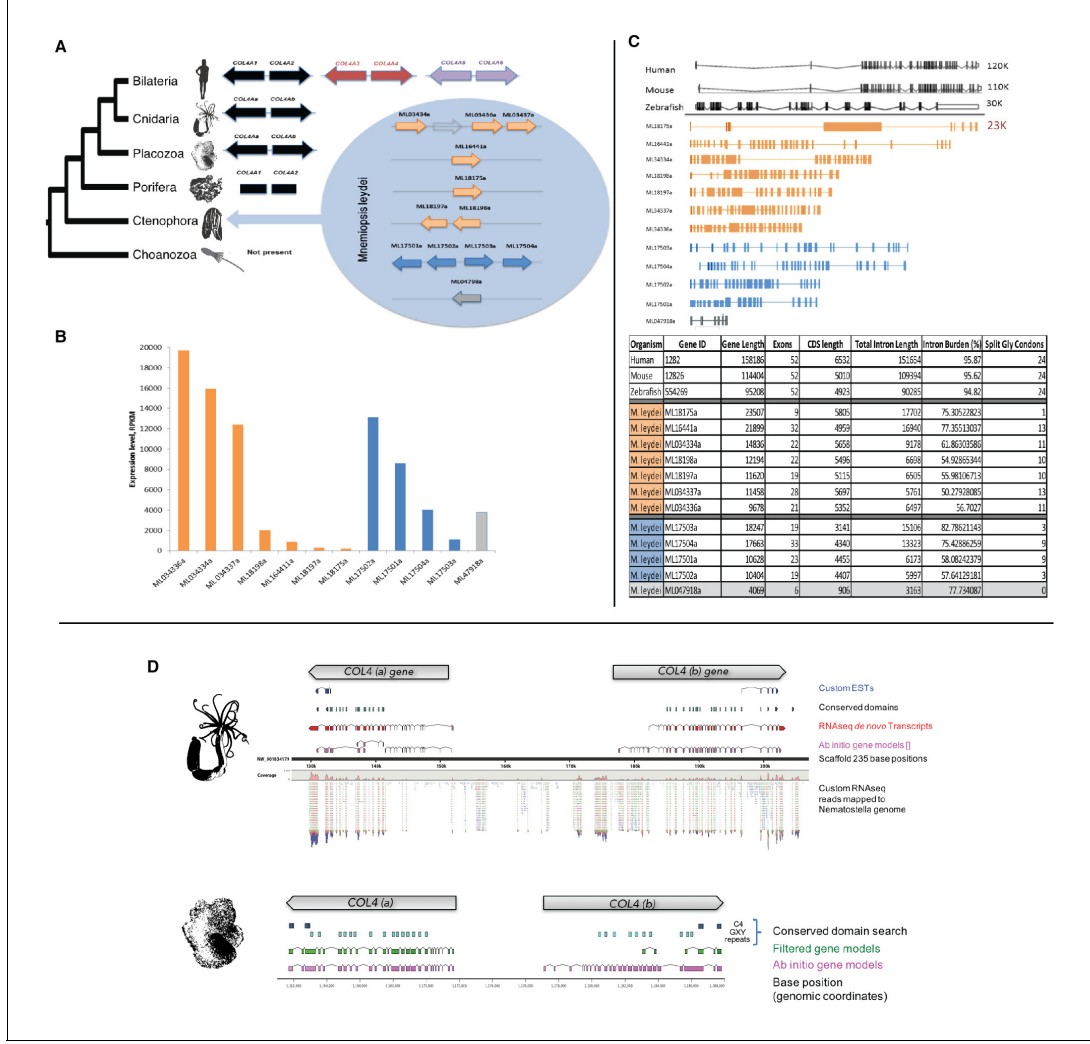

**Figure 3.** *Mnemiopsis* reveals multiple duplications of collagen IV genes, and non-bilaterian animal phyla collagen IV organization is similar to Bilateria. (**A**) Collagen IV Gene Orientation. *Mnemiopsis* collagen IV genes were separated into two groups base on genomic orientation. Group I genes are found on the same scaffold (colored in blue). Group II genes (colored in orange) are spread across four different scaffolds and do not have head-to-head orientation. (**B**) Transcriptome analysis of *Mnemiopsis* confirmed a total of 11 collagen IV genes and one NC1 proto-domain gene (colored in grey). (**C**) Human COL4A1 spans 150 kb, contains 52 exons and has an intron composition of 95%. *Mnemiopsis* collagen IV genes are approximately one sixth the length of human collagen IV genes ranging in length from 1 to 23 kb with an intronic composition of 50–82%. (**D**) In *Nematostella* and *Trichoplax,* two collagen IV genes are located in head-to-head orientation on one genomic scaffold (*Nematostella*, scaffold 14; *Trichoplax*, scaffold 235), which indicates that they share one chromosome. The arrows at the top of each species indicate gene orientation: either *minus* or *plus* strands. The search for conserved domains revealed multiple collagens repeats (PF01391, light blue boxes) and C4 domains (PF01413, dark blue) further support that these genes belongs to collagen IV gene family. Pfam domains were identified using HMM against the genomic sequence. Mapping RNAseq reads to the genome strongly supports the proposed collagen IV genes model.

The following source data and figure supplements are available for figure 3:

**Source data 1.** *Mnemiopsis* collagen IV gene expression by RNA-Seq (RPKM).

**Figure supplement 1.** In an effort to detect conservation of exon size between ML collagen IV genes a database composed of all the exons from each gene was compiled.

**Figure supplement 1—source data 1.** Frequency of >exon lengths in *Mnemiopsis* collagen IV genes.

**Figure supplement 1—source data 2.** Conservation of exon length and position is present among some of the groups.

*Figure 3 continued*

**Figure supplement 2.** The first NC1 domain (red) coding exon of collagen IV has several features conserved throughout the animal kingdom, including the collagenous domain (yellow), which is a stretch of interrupted Gly-X-Y repeats at the 5′ end, and the presence of an HSQ coding region (white text in red).

(*Figure 3A*). Furthermore, ML17502a and ML17503a genes are oriented in opposite directions, separated by ~2879 bases, and do not share the same promoter. The other seven *Mnemiopsis* genes: ML166441a, ML18175a, ML18197a, ML18198a, ML034334a, ML0343336a, and ML0343337a (designated as Group II), are aligned individually on separate scaffolds or in a unidirectional tandem array. Transcriptomic analysis of adult *Mnemiopsis* reveals differential expression within each group of collagen IV genes. In Group I, the expression of ML17501a and ML17502a is significantly higher than other Group I genes. Similarly, ML034334a, ML0343336a, and ML0343337a show considerable increase in expression levels compared to the other Group II genes (*Figure 3B*).

*Mnemiopsis* collagen IV genes are shorter in sequence length and typically have fewer exons and shorter intronic regions compared to their Bilaterian counterparts (*Figure 3C*). Whereas bilaterian fibrillar collagen genes are composed of multiple exons with a length of 54 base pairs (*Yamada et al., 1980*), exons of *Mnemiopsis* collagen IV genes range in size from 40 to 4867 bp. Among the *Mnemiopsis* genes, exons of 108 bp in length were found in several genes, including ML17503a, ML17504a, and ML16441a, but no increased frequency of 54 bp exons in length or any variation was detected) (*Figure 3—figure supplement 1*). The presence of split glycine codons coding for the collagenous domain and codons on junctional exons (e.g. collagenous domain/NC1 domain encoding exons) are defining features of vertebrate collagen IV genes (*Quinones et al., 1992*). Multiple genes from each group (Group I: ML17501a, ML17502a, and ML17503a; Group II: ML034334a, ML034336a, and ML034337a) possess split glycine codons on the collagenous domain/NC1 domain junction exon (*Figure 3—figure supplement 2* and *Supplementary file 1*).

We then determined the number of collagen IV genes in 10 other species from the Ctenophora phylum. Transcriptome analysis was conducted using in-house generated libraries for *Mnemiopsis leidyi* and *Pleurobrachia pileus* and publicly available libraries for *Pleurobrachia bachei, Beroe ovata, Beroe abyssicola, Euplokamis sp., Dryodora sp., Vallicula sp., Coeloplana sp.,* and *Bolinopsis sp.* Across these ten species, a total of 118 unique collagen IV genes were detected. Each species contained a variable number of collagen IV genes, ranging between 4 and 20 genes each, as compared to the 2 to 6 genes in the other non-bilaterian animal phyla and Bilateria (*Figure 4A and B*). All species, apart from the two *Beroe* species surveyed, contain a combination of both Group I and II chains, with the two *Beroe* species exhibiting only Group II chains. In addition to full-length collagen IV genes, two genes, with signal peptides, encoding only the NC1 domain were identified across ten species of Ctenophora. Expression of NC1 domains without collagenous tails is novel (*Figure 4C*). Together, these findings show that the ECM of ctenophores contain both collagen IV and standalone NC1 genes, and that the number and diversity collagen IV genes exceeds that of any other metazoan group.

We also determined the number, structure, and orientation of collagen IV genes in *Nematostella vectensis* (Cnidaria) and *Trichoplax adhaerens* (Placozoa). We found two genes in both species, and that they are homologous with Bilateria, with each demonstrating a 'head-to-head' orientation and homologous coding regions for both collagenous and non-collagenous domains (*Figure 3D*). Complete genomic data was unavailable to determine the orientation of the two collagen IV genes in homoscleromorph sponges. In contrast, unicellular protists (Choanozoa, Filasterea, Amoebozoa, Apusozoa) do not contain collagen IV as determined by genomic analyses (*Figure 2*). Together, our findings show that head-to-head orientation of collagen IV is conserved across Bilateria, Cnidaria, Placozoa, and Ctenophora, whereas Ctenophora exhibits both head-to-head and tandem orientations (*Figure 3A*).

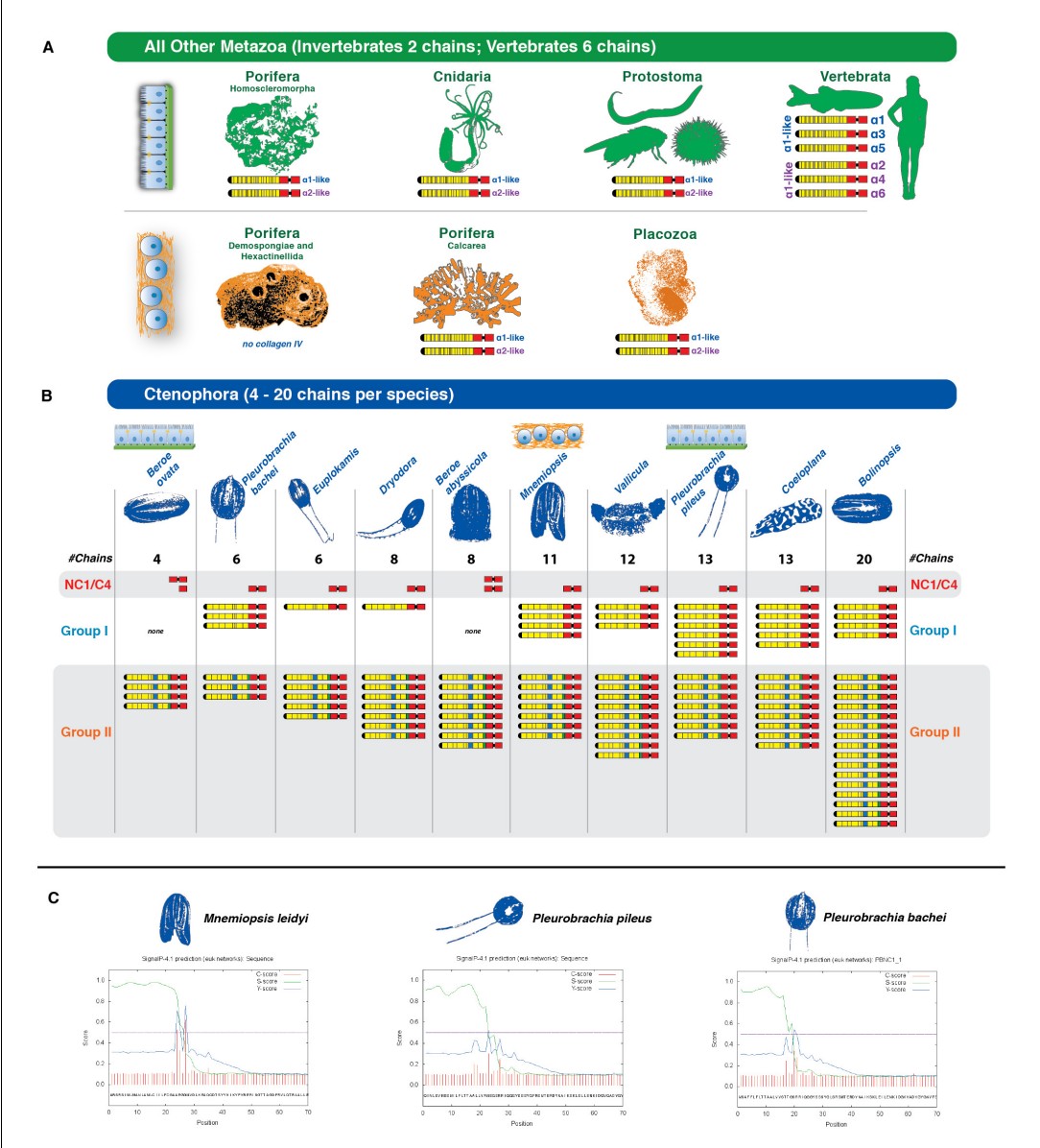

**Figure 4.** Collagen IV in Ctenophora underwent numerous gene duplication events resulting in an unprecedented diversity. (**A**) Collagen IV chain distribution across non-bilaterian animal phyla and Bilateria. Two collagen IV chains are found across invertebrates, and six chains in chordate/vertebrate lineages. The poriferan class of Demosponges lacks collagen IV and BM. (**B**) Ctenophora collagen IV chains range from four to twenty distinct chains across species, indicating a variable number of gene duplication events. Ctenophora chains can be split into Group I, Group II, and NC1/C4 subgroupings. All ctenophore species contain Group I, II, and NC1 genes except for the two *Beroe* species, which lack Group I chains. (**C**) NC1 genes identified across Ctenophora were analyzed for signal peptide presence to determine whether sequences were truncated, or represented standalone NC1 proteins. Putative signal peptides were detected in at least three ctenophore NC1 genes, *Mnemiopsis* (ml047918a), *Pleurobrachia pileus* (pp_COL4_i), and *Pleurobrachia bachei* (PBNC1_1) based on SignalP prediction (http://www.cbs.dtu.dk/services/SignalP/).

## Structural domains of non-Bilaterian animal phyla collagen IV are homologous to Bilaterian ones

Several prominent structural domains characterize Bilaterian collagen IV chains (*Figure 5*). These include an N-terminal non-collagenous domain rich in cysteine and lysine residues (NC3) (*Figure 5—figure supplement 1*), a large collagenous domain of Gly-Xaa-Yaa (GXY) repeats of ~1400 residues with interruptions in the GXY repeats (*Figure 6*), followed by a non-collagenous (NC1) domain at the

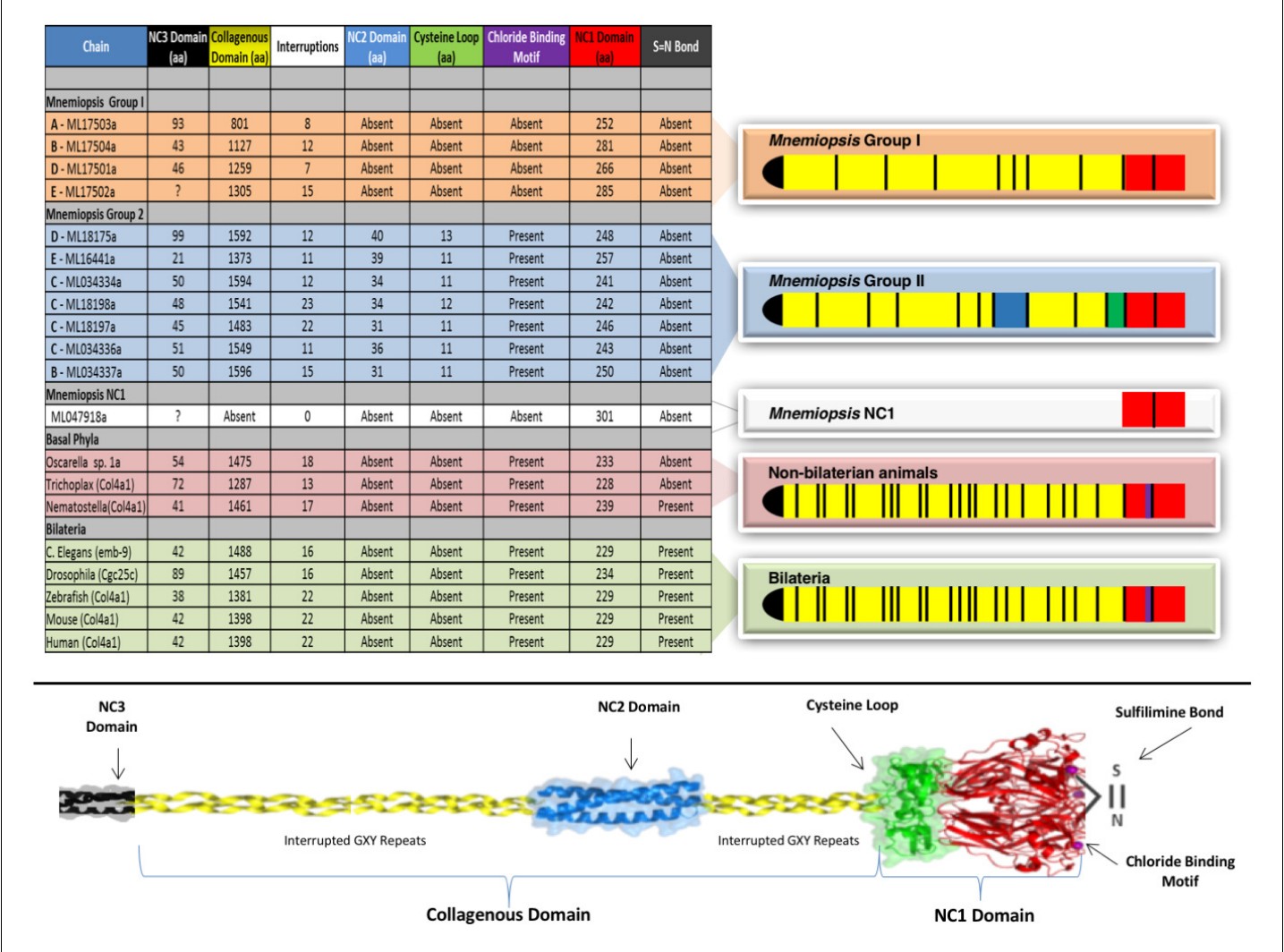

**Figure 5.** Collagen IV structural features are conserved across metazoa, and Ctenophora exhibits novel domains. Signature features of collagen IV are found in each of the identified in *Mnemiopsis* chains. The collagenous region (yellow) of each chain contains characteristic interruptions (black lines) of the Gly-X-Y motif repeats. Group II chains also possess a NC2 domain (blue), which interrupts the collagenous region, and a cysteine loop (green) that is an extension of canonical NC1 domain (red). The NC1 domain of each chain is composed of two C4 domains. Group II chains possess the chloride-binding motif (purple) within the NC1 domain. While conservation of most *Mnemiopsis* collagen IV features can be found throughout metazoan species the NC2 domain and cysteine loop are structural innovations restricted to Ctenophora.

The following figure supplements are available for figure 5:

**Figure supplement 1.** Multiple sequence alignment of the NC3 domain from various metazoan species reveals a high degree of conservation in this region among *Mnemiopsis* sequences, which is not seen in other metazoan sequences.

**Figure supplement 2.** Multiple-sequence alignment of collagen IV NC1 domain sequences across human, mouse, zebrafish, fly, *C. elegans*, *Nematostella*, and *Trichoplax,* compared to the ctenophore representative, *Mnemiopsis leidyi* (MLXXXXXX).

**Figure supplement 3.** The chloride-motif has been identified previously in humans to *Trichoplax*.

**Figure supplement 4.** Sulfilimine bond crosslinking of collagen IV occurs between Methionine-93 and Lysine/Hydroxylysine-211 residues between adjoining NC1 domain interfaces.

**Figure supplement 5.** Multiple sequence alignment of 32 partial ctenophore collagen IV sequences spanning 10 species reveals the NC2 domain (*highlighted in blue*), spanning 38–44 amino acids.

*Figure 5 continued on next page*

*Figure 5 continued*

**Figure supplement 6.** Multiple sequence alignment of partial sequences from 41 collagen IV genes exhibiting the cysteine-loop region of NC1 domains in Group II chains across Ctenophora.

C-terminus of approximately ~230 residues (*Figure 5—figure supplement 2*). NC1 domains are comprised of two C4 domains, each containing a short, highly conserved HSQ residue motif proximal to the N-terminal side. We sought to determine whether these structural domains are characteristic of Ctenophora and the other non-bilaterian animal phyla. Indeed, these domains are conserved across Cnidaria, Placozoa, Porifera, and Ctenophora (*Figure 5*). The NC1 domain also contains a chloride-binding motif, which functions in binding extracellular chloride to signal the assembly of collagen IV networks and is conserved from vertebrates to Cnidaria and Placozoa (*Cummings et al., 2016*). The chloride-binding motif was identified in both Ctenophora (group II chains) as well as in the two Homoscleromorph sponges analyzed, suggesting the chloride signaling function of NC1 domains is also conserved in Ctenophora and Porifera (*Figure 5—figure supplement 3*). Together, these findings reveal that the conserved structural features of bilaterian collagen IV extend across the non-bilaterian animal phyla, including Ctenophora.

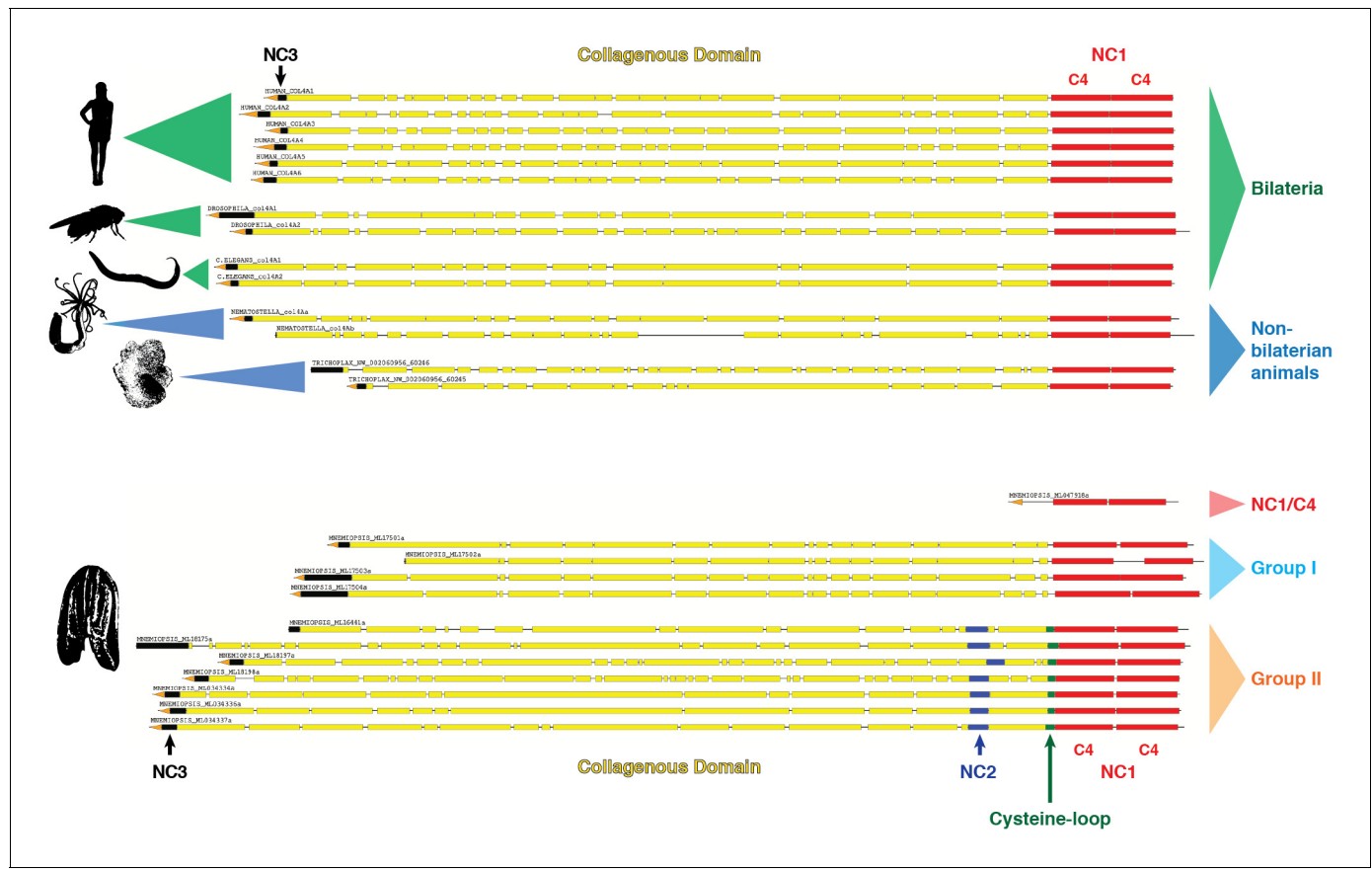

**Figure 6.** Collagen IV in Ctenophora and the non-bilaterian animal phyla are structurally homologous to bilateria. Representative collagen IV sequences from human (UniProt entries P02462, P08572, Q01955, P53420, P29400, Q14031), *Drosophila* (UniProt entries P08120, O18407), *C. elegans* (UniProt entries P17139, P17140), *Nematostella*, *Trichoplax*, and *Mnemiopsis* genomes. The following regions are depicted: (based on prediction from http://www.cbs.dtu.dk/services/SignalP/; shown as orange arrow), NC3 domain (black box), uninterrupted triple helical segments with at least three GXY repeats (yellow boxes), NC2 domain (blue box), cysteine loop (green box), C4 domains (based on conserved domain search at http://www.ncbi.nlm.nih.gov/Structure/cdd/wrpsb.cgi; shown as red boxes).

## The ctenophore NC1 recognition module harbors a novel crosslink

The NC1 domain is the molecular recognition module that directs the assembly of collagen IV protomers and networks (*Cummings et al., 2016*). NC1 modules function in selecting collagen IV chains for trimerization, forming triple-helical protomers, and for oligomerization of protomers into networks (*Figure 5*) (*Cummings et al., 2016*; *Khoshnoodi et al., 2008*). The NC1 modules are stabilized by sulfilimine cross-links, which connect methionine-93 (Met-93) and lysine/hydroxylysine-211 (Lys/Hyl-211) between adjoining protomers (*Fidler et al., 2014*; *Vanacore et al., 2009*). Sulfilimine cross-links are conserved throughout Bilateria, from Humans to *C. elegans*, and in Cnidaria, with the exception of *Hydra* (*Fidler et al., 2014*). In contrast, this cross-link is absent in Ctenophora, owing to the absence of Met-93 and Lys/Hyl-211 residues (*Figure 5—figure supplement 4*). Thus, we sought to characterize the biochemical properties of ctenophore NC1 domains to ascertain whether they are stabilized by an alternative cross-linking mechanism. Uniquely, ctenophore collagen IV is distinguished from all other metazoans by the presence of two additional domains. One is a non-collagenous domain (NC2 domain) that is approximately 38–44 residues in length within the collagenous domain (*Figure 5—figure supplement 5*). The other domain is 11–13 residues in length and consists of 3–4 conserved cysteine residues, designated as the cysteine-loop, which is an extension of the canonical NC1 domain, and a candidate for an alternative cross-linking mechanism (*Figure 5—figure supplement 6*). As we previously established for bilaterian collagen IV, the presence of NC1 dimers after reduction of with mercaptoethanol, indicates cross-links (*Fidler et al., 2014*; *Vanacore et al., 2009*). Analysis in the three ctenophore species, *Mnemiopsis*, *Beroe* and *Pleurobrachia*, by SDS-PAGE and gel filtration chromatography, revealed the presence of NC1 hexamers, which upon reduction dissociated into dimers (*Figure 7A–D*). Since the dimers lack Met-93 and Lys/Hyl-211, the results indicate that ctenophore dimers are stabilized by the cysteine-loop (*Figure 7E*). Hence, the cross-linking mechanism of ctenophores (cysteine-loop) is distinguished from that of Cnidaria and Bilateria (sulfilimine cross-links).

## Ctenophore collagen IV NC1 recognition modules are more diverse and distinct than all other metazoans

NC1 domains are distinguishing domains of collagen IV that function as recognition modules in the assembly of collagen IV networks (*Cummings et al., 2016*). Uniquely, all 10 ctenophore species possess a gene encoding only the NC1 domain, a feature not found in any other phyla. We performed phylogenetic analysis to compare the NC1 domains of non-bilaterian animal phyla with that of Bilateria (*Figure 8*). The results placed the NC1 domain of ctenophore collagen IV into two major groups, which are consistent with genomic orientation of Group I and Group II (*vida supra*). We conducted additional phylogenetic analysis of the NC1 domain using RAxML to select the best tree from eleven models of evolution (DAYHOFF, DCMUT, JTT, MTREV, WAG, RTREV, CPREV, VT, BLOSUM62, MTMAM, and LG). Among these, the VT model yielded the tree with the best-fit (*Figure 8—figure supplement 1*). The distinction of the two groups coincides with the presence of the novel structural domains, NC2 and cysteine-loop, found exclusively in ctenophore Group II chains. Furthermore, these groups can be further subdivided into subgroups Group I (A-E) and Group II (A-D) based on phylogenetic affinity (*Figure 8—figure supplement 1*). RAxML phylogenetic analysis revealed a closer affinity between the ctenophore NC1 genes and collagen IV NC1 domain from the non-bilaterian animal phyla and Bilateria, as compared to ctenophore Group I and II chains (*Figure 8*). Furthermore, Group I and Group II ctenophore chains showed a much higher rate of divergence both within the phylum and as compared to the other non-bilaterian animal phyla and bilaterian collagen IV sequences. The unrooted tree topology illustrates the high phylogenetic affinity between ctenophore NC1 proteins and bilaterian NC1 domain that cluster closely.

## Spongin short chain collagens in Porifera, Placozoa, and Cnidaria are collagen IV variants

Spongins are a family of collagen IV-related proteins composed of a short collagenous domain attached to an NC1 domain. This protein family was first detected in the exoskeleton of demosponges and has been subsequently identified in cnidarians, across invertebrates (with the exception of ecdysozoans, e.g., *C. elegans*, *Drosophila*), and in basal chordates (*Aouacheria et al., 2006*; *Exposito et al., 1991*). Interestingly, spongins do not occur in vertebrates (*Aouacheria et al.,*

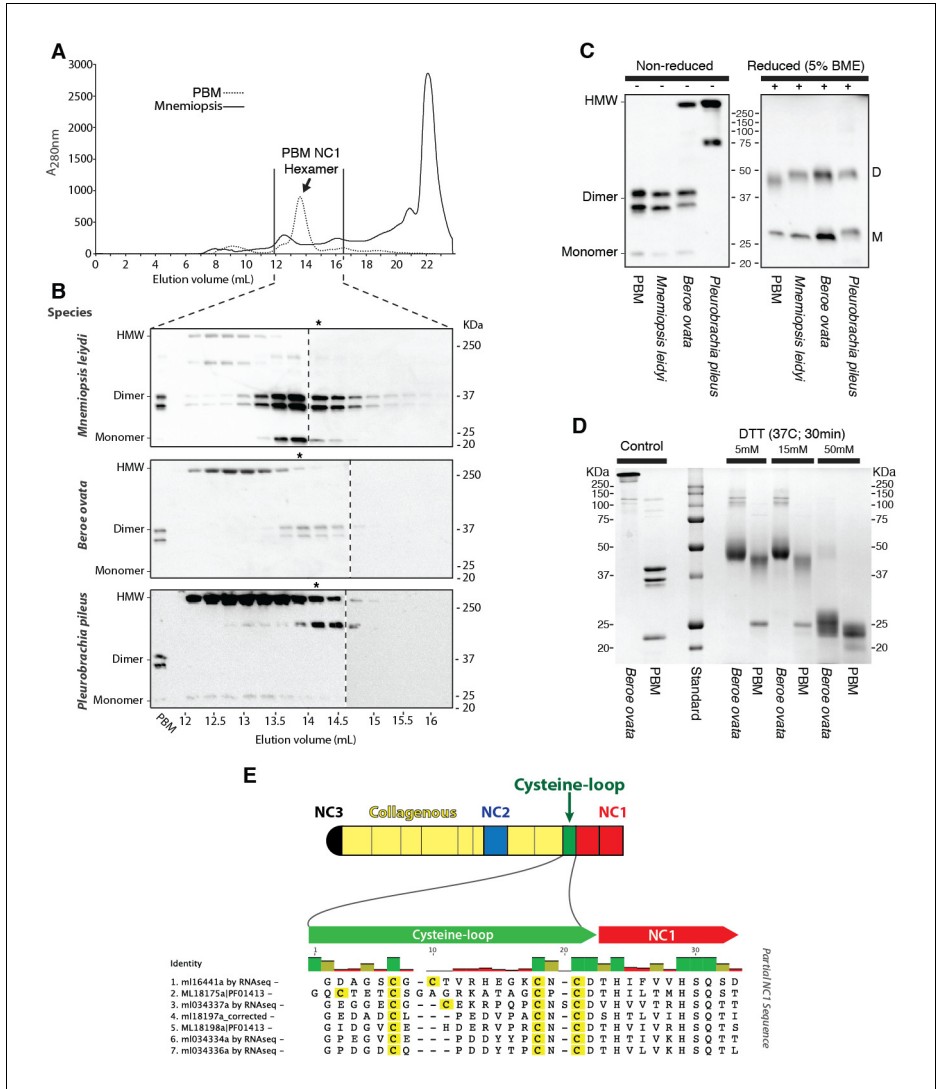

**Figure 7.** Ctenophora exhibits a novel collagen IV cross-linking mechanism. (**A**) Gel filtration chromatography elution profile of *Mnemiopsis* collagenase digest (black) and native, purified placental basement membrane NC1 hexamer (dashed) run successively. Three ctenophore species were digested with bacterial collagenase to solubilize NC1 hexamer for analysis of collagen IV crosslinking. (**B**) Western blot of gel filtration fractions encompassing elution of NC1 hexamer (12 mL to 16.4 mL) from *Mnemiopsis, Pleurobrachia,* and *Beroe*, developed with NC1-specific monoclonal antibodies. HMW=high-molecular-weight complex. (**C**) Western blot of ctenophore NC1 hexamer separated by SDS-PAGE under reducing (+) and non-reducing (-) conditions (5% *β*-mercaptoethanol). (**D**) Reduction of the high-molecular-weight complex from *Beroe* (first lane, >250 kDa) following by alkylation results in formation of dimers at low DTT concentration, and complete reduction to monomers at high DTT concentration. (**E**) Structure of Ctenophora collagen IV group II chain, highlighting cysteine-loop region of the NC1, and multiple-sequence alignment of cysteine-loop region of Group II chains of *Mnemiopsis* (NC1 domain is partial sequence).

2006), and we did not detect them in Ctenophora. We examined the phylogenetic relationship of the NC1 domain of spongins to that of collagen IV NC1 domains (*Figure 8* and *Figure 8—figure supplement 1*). Multiple sequence alignment showed conservation of seven cysteine residues between spongins and collagen IV, while the HSQ motif was absent in spongin sequences (*Figure 9*). Comparison of collagen IV sequences revealed four cysteine residues that are absent in spongin sequences. The spongin variants, however, do show conservation of three cysteines that are absent in collagen IV sequences. Collectively, the presence of collagenous domains and the conservation of

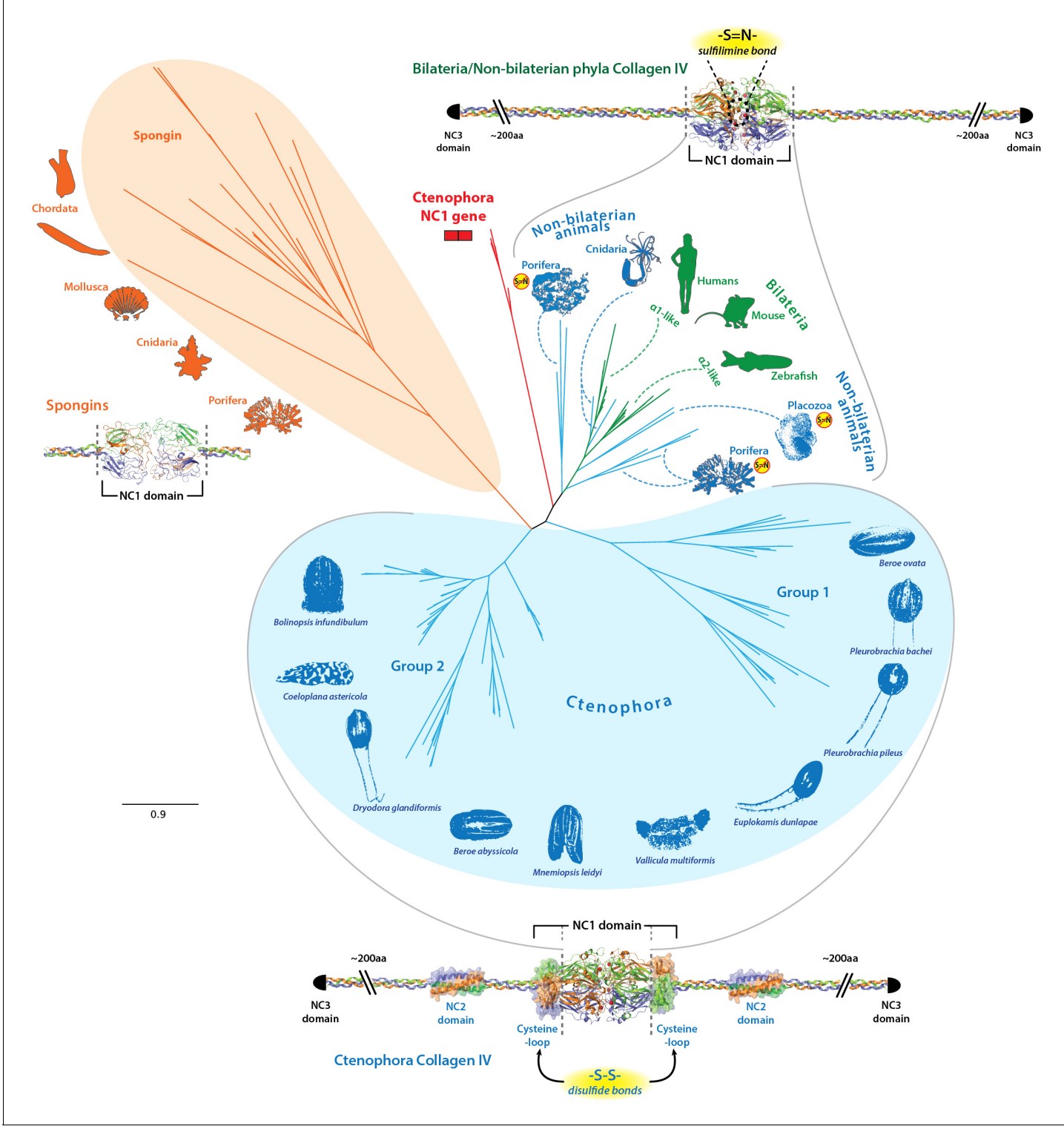

**Figure 8.** Evolutionary relationships of collagen IV and spongin NC1 domains across metazoa compared to Ctenophora. Metazoan collagen IV chains feature a sulfilimine bond cross-linked collagen IV network, with the exception of the cnidarian, *Hydra*, and the non-bilaterian animal phyla Porifera and Placozoa. However, the structural domains across bilaterians and the non-bilaterian animal phyla are homologous; however, Ctenophora also contains novel domains. Unrooted maximum likelihood tree of collagen IV NC1 domains in human, mouse, zebrafish, *Trichoplax* (Placozoa), *Pseudocorticium jarrei* (Porifera), and *Oscarella sp.* (Porifera), in comparison with 10 ctenophore species. All analyses were based off amino acid sequence alignments of the NC1 domain, omitting the cysteine-loop region of Ctenophora NC1 domains.

*Figure 8 continued on next page*

*Figure 8 continued*

The following figure supplement is available for figure 8:

**Figure supplement 1.** NC1 domain phylogeny across metazoa.

collagen IV NC1 domain features within spongin NC1 domains, reveal that they are homologous to collagen IV protein domain structure, as previously noted (*Exposito et al., 1991*).

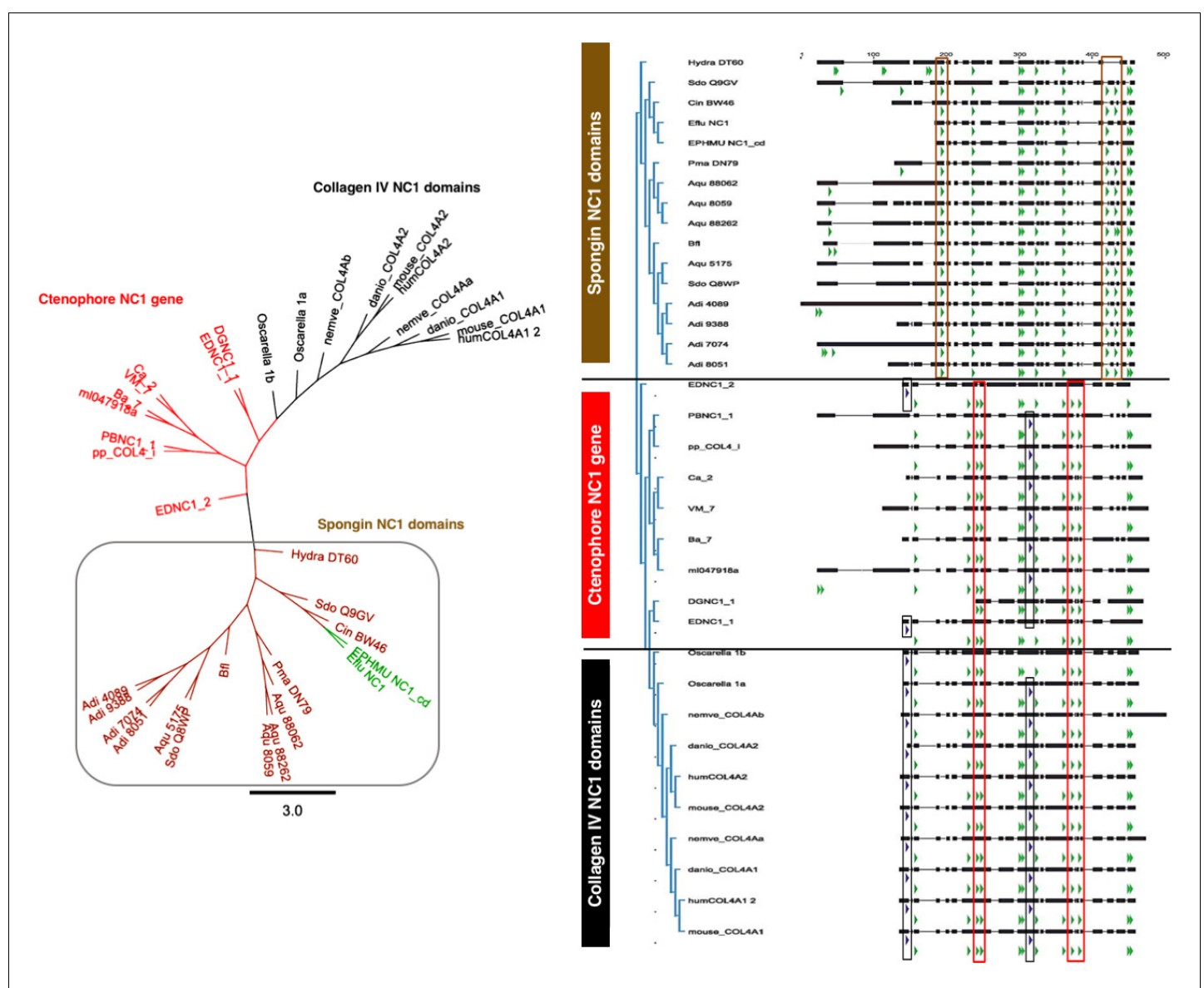

**Figure 9.** Spongins show conservation of key primary structure features within the NC1 region as compared to collagen IV. Multiple sequence alignment reveals the conservation of cysteine residues (green arrows) across all four families of collagen IV. Seven cysteines are common to all sequences, while spongins share three unique cysteine residues (brown box). Likewise, the ctenophore NC1 protein and the NC1 domain of bilaterian collagen IV show conservation of four cysteine residues not found in spongin sequences (red box). The ctenophore sequences also show conservation of the second HSQ motif (purple arrows) found within the bilaterian NC1 domain (black box). No HSQ motifs were detected in the spongin sequences.

## Discussion

Within Ctenophora, collagen IV underwent numerous gene duplication events resulting in an unprecedented diversity in both gene sequence and organization in comparison to all other metazoans. Ctenophora contains between 4 and 20 collagen IV chains across species, and exhibits both head-to-head orientation and genes aligned individually on separate scaffolds or in a unidirectional tandem array. Interestingly, Ctenophora has both collagen IV genes and a standalone NC1 gene. In both *Nematostella* and *Trichoplax,* there are two collagen IV genes exhibiting head-to-head gene orientation, similar to that of Bilateria. In Porifera, the ECM of homoscleromorphs is composed of two collagen IV genes, while in demosponges, it is composed of spongin, a collagen IV variant. Spongin is absent in Ctenophora but is present in non-bilaterian animal phyla, invertebrates, and lower chordates along with collagen IV throughout invertebrates and lower chordates, with the exception of *Drosophila* and *C. elegans.* Structural and phylogenetic analysis of spongin shows it is homologous to collagen IV (*Figures 8* and *9*). Collectively, collagen IV genes are highly conserved across the non-bilaterian animal phyla and Bilateria; yet, in Ctenophora, these genes are more diverse and distinct including a novel cross-linking mechanism, with up to 20 distinct genes compared with six in vertebrates. Moreover, the collagen IV gene is absent in the unicellular sister-groups (choanoflagellates, filastereans, amoebozoans, and apusozoans), suggesting it was an early metazoan innovation.

To address the evolutionary origin of the collagen IV gene, we compared two scenarios, Ctenophora-first versus Porifera-first (*Ryan et al., 2013*; *Moroz et al., 2014*; *Whelan et al., 2015*; *Telford et al., 2016*). In both scenarios, collagen IV appeared in an early metazoan ancestor or a unicellular ancestor but was secondarily lost in demosponges and hexactinellid sponges (*Figure 10A and B*). It is noteworthy that the NC1 gene is present only in Ctenophora (*Figure 10C and D*), suggesting that this gene is a remnant from an early metazoan ancestor, and a forerunner to the NC1 domain of the ancestral collagen IV gene. This NC1 gene encodes a key recognition module that directs the assembly of collagen IV suprastructures (*Cummings et al., 2016*). In comparison, spongin appeared after the divergence of the Ctenophora phylum in the Ctenophora-first hypothesis, yet in the Porifera-first hypothesis it would have appeared alongside collagen IV in the early metazoan ancestor or unicellular ancestor. With either hypothesis, the collagen IV gene coincided with the appearance of multicellular animals. Although laminin genes appear to have arisen prior to the metazoan lineage, with laminin-related genes appearing in unicellular choanoflagellates (*Fahey and Degnan, 2012*) (*Figure 10A and B*).

Collectively, we propose a model for collagen IV gene evolution that incorporates both Ctenophora-first and Porifera-first hypotheses (*Figure 11*). The presence of Gly-X-Y collagenous repeats, in the absence of a collagen IV gene, in choanoflagellates and amoebozoa, and the presence of a NC1 gene in the Ctenophora phylum suggests that Gly-X-Y repeats combined with an NC1 domain gene in an early metazoan ancestor, or possibly in a unicellular ancestor, forming an ancestral collagen IV gene. This combination of domains is analogous to the domain shuffling events that gave rise to the developmental protein, *hedgehog* (*Adamska et al., 2007*). Within Ctenophora, collagen IV genes underwent unprecedented experimentation with several duplication events resulting in up to twenty distinct genes with both tandem and head-to-head organization. Within Porifera, the collagen IV gene was duplicated with a head-to-head orientation. This head-to-head feature was conserved in both sequence and gene structure throughout non-bilaterian animals and Bilateria (*Figure 11*), with the known exception of *C. elegans* (*Guo and Kramer, 1989*). Two additional rounds of genome duplication resulted in six collagen IV genes in the vertebrate subphylum. Spongin, a collagen IV variant, first appeared in Porifera and is conserved throughout invertebrates, with the exception of Ecdysozoa and is found in cephalochordates (*Branchiostoma floridae*) and tunicates (*Ciona intestinalis*). The spongin gene arose either by domain shuffling of Gly-X-Y repeats from a unicellular ancestor and the NC1 gene, analogous to the assembly of the ancestral collagen IV gene, or diverged from an ancestral collagen IV gene (*Figure 11*).

Collagen IV protein, or its spongin variant, is a required ECM component for all extant multicellular animals, considering that all animals investigated contain either collagen IV or spongin, and that the essentiality of collagen IV during development has been established in several studies (*Gupta et al., 1997*; *Pöschl et al., 2004*; *Gotenstein et al., 2010*; *Bhave et al., 2012*). The collagen IV protein is associated with two distinct organizations of cells; one in which the ECM contains

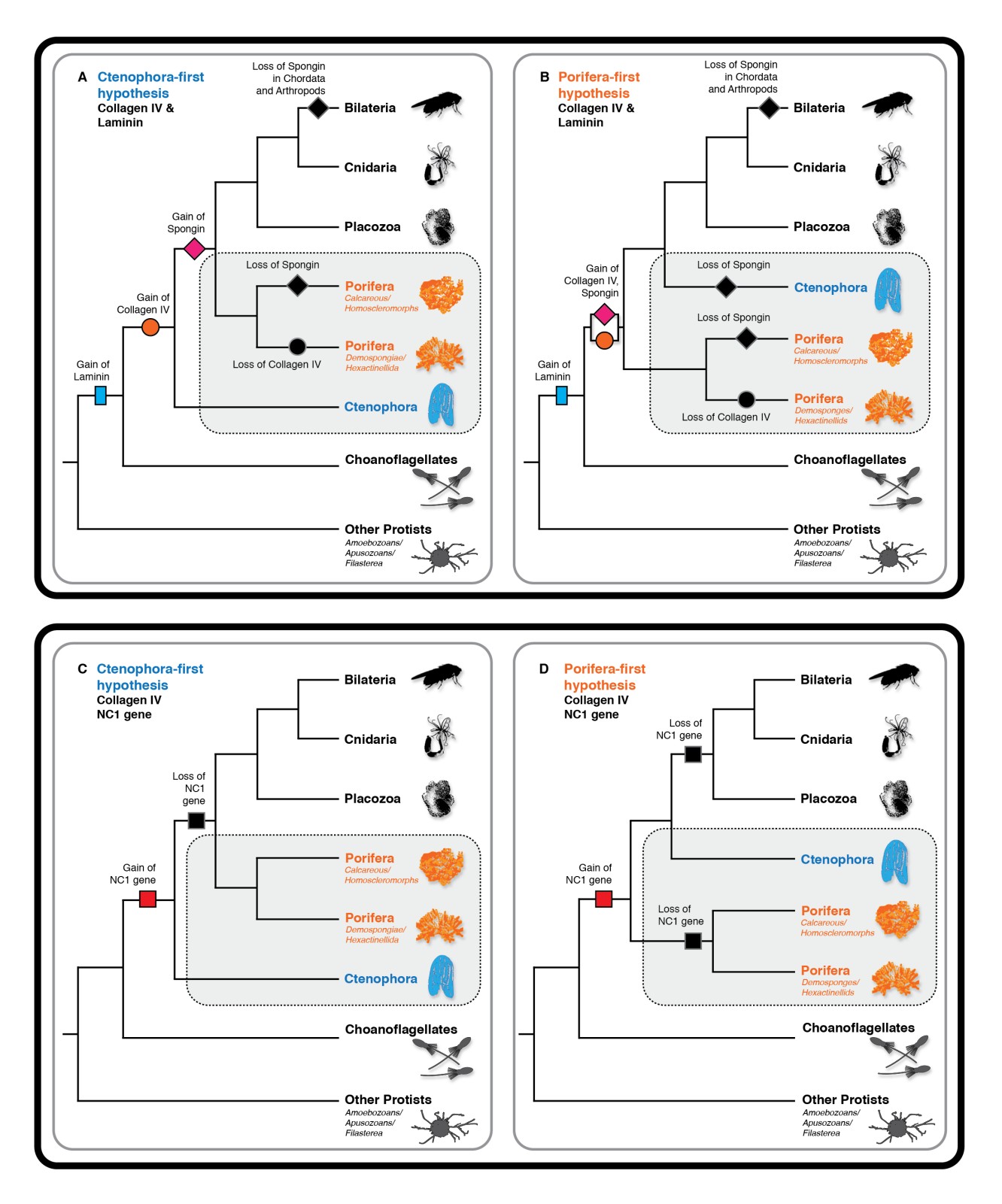

**Figure 10.** Collagen IV and Laminin gene evolution under Ctenophora or Porifera-first hypotheses. (**A** and **B**) Comparison of collagen IV, spongin, and laminin gene evolution gain and loss evolutionary events in Ctenophora-first and Porifera-first hypotheses. (**C** and **D**) Comparison of NC1 gene evolution gain and loss evolutionary events in Ctenophora-first and Porifera first hypotheses.

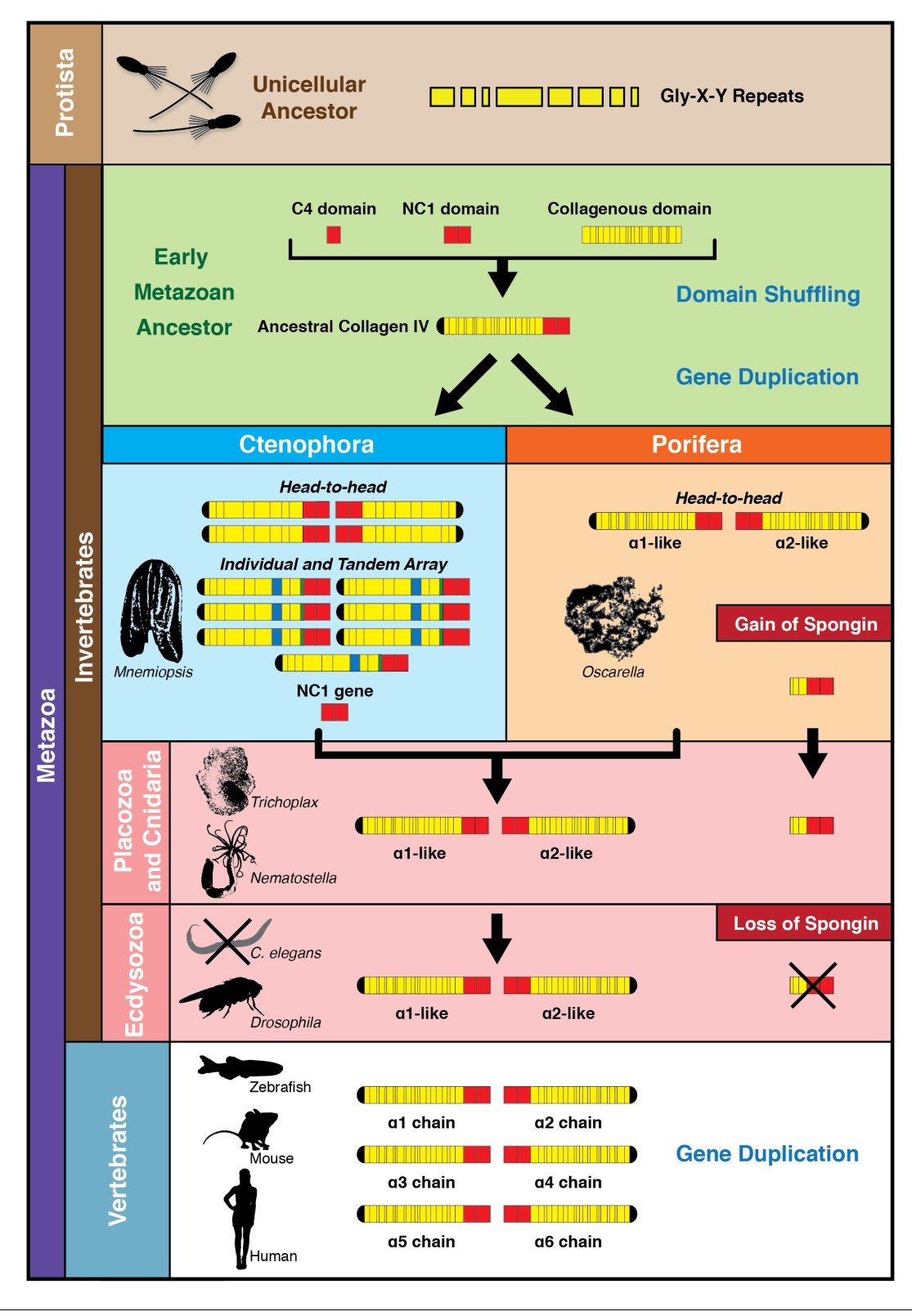

**Figure 11.** The non-bilaterian animal phyla reveal an evolutionary model for collagen IV. Based off our phylogenetic analysis: (1) the presence of the ancestral NC1 domain in Ctenophora may have resulted from tandem duplication of the ancestral C4 or conservation of the ancestral NC1 domain. The last common ancestor of ctenophore and the non-bilaterian animal phyla may have expressed both the ancestral C4 and the ancestral NC1 domain. (2) Intergenic duplication of the ancestral collagen IV resulted in the head-to-head orientation. Errors in gene duplication may have given rise to spongins

*Figure 11 continued on next page*

*Figure 11 continued*

as they lack the domain-swapping region of the NC1 domain (a determined by predicative modeling) and have truncated collagenous tails. (3) The presence of six collagen IV genes arranged in a head-to-head orientation in vertebrates likely resulted from the two rounds of genome duplication that occurred in the vertebrate lineage.

collagen IV broadly dispersed between communities of cells (*Mnemiopsis*) and the other in which collagen IV is a component of a well-defined BM underlying a layer of cells (*Beroe*, *Pleurobrachia*, Homoscleromorph sponges, and *Nematostella*), a hallmark feature of epithelial bilaterian tissues (*Figure 12*). The absence of BMs in *Trichoplax* and *Mnemiopsis* suggests there is an unknown component that facilitates the assembly of collagen IV and laminin into a basement membrane.

Collectively, we conclude that collagen IV and its spongin variant are primordial components of the extracellular microenvironment, and collagen IV, as a component of BM, enabled the assembly of a fundamental architectural unit for the genesis and evolution of multicellular tissues (*Figure 12*). This unit is characterized by a layer of apical/basal-polarized cells that are laterally connected by

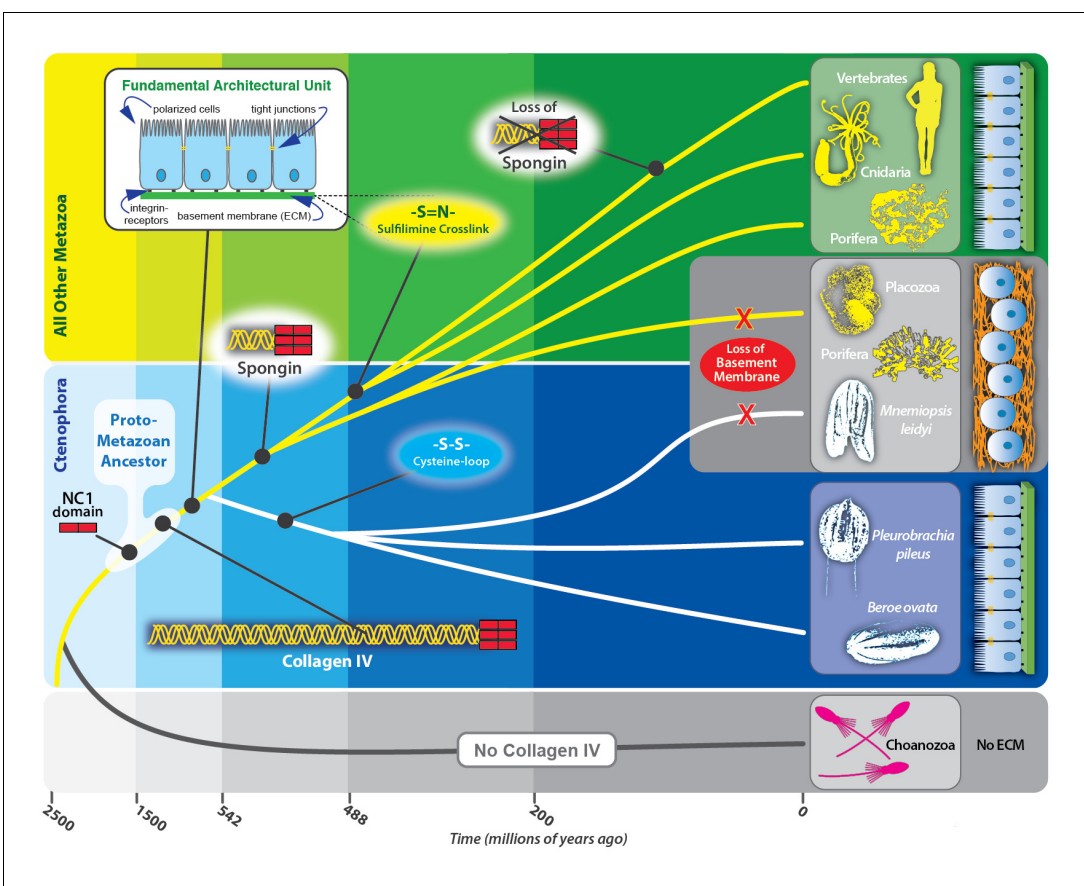

**Figure 12.** Collagen IV enabled the transition to multicellularity and the evolution of epithelial tissues in metazoa. Collagen IV was a primordial innovation in early metazoan evolution, providing the architectural foundation for ECM formation. Choanoflagellates exist as singular or in colonies, yet do not have an ECM. Spongins are similar in domain structure and phylogeny to NC1 domains across metazoan collagen IV, and are variants of collagen IV, arising during the divergence of demosponges. Collagen IV, as a member of the basement membrane toolkit, enabled the evolution of multicellularity. Basement membranes juxtaposed to plasma membrane underlying a layer of polarized cells are a fundamental architectural unit of epithelial tissues. A layer of apical/basal-polarized cells that are laterally connected by tight junctions between plasma membranes, and basally anchored via integrin receptors embedded in plasma membranes to a basement membrane suprastructure is a fundamental architectural unit.

tight junctions between plasma membranes, which are basally anchored via integrin receptors embedded in plasma membranes to a basement membrane supra-scaffold. In turn, this architectural unit served as the building block that enabled the formation and evolution of epithelial tissues, the ever-increasing complexity and size of organisms, and for the expansion and diversity of the animal kingdom.

## Materials and methods

### Next-generation RNA sequencing (RNA-Seq)

Transcriptomes used in this study were sequenced at the Vanderbilt Technologies for Advanced Genomics Core Facility (VANTAGE, Nashville, TN). The Illumina TruSeq mRNA Sample Preparation Kit was used to convert the mRNA in 100 ng of total RNA into a library of template molecules suitable for subsequent cluster generation and sequencing on the Illumina HiSeq 2500 using the rapid run setting. The pipeline established in VANTAGE was followed and is briefly described below. The first step was a quality check of the input total RNA by running an aliquot on the Agilent Bioanalyzer to confirm RNA integrity. The Qubit RNA fluorometry assay was used to measure sample concentrations. The input-to-library prep was 100 ng of total RNA (2 ng/ul). The poly-A containing mRNA molecules were concentrated using poly-T oligo-attached magnetic beads. Following purification, the eluted poly(A) RNA was cleaved into small fragments of 120–210 base pair (bp) using divalent cations under elevated temperature. The cleaved RNA fragments were copied into first strand cDNA using SuperScript II reverse transcriptase and random primers. This step was followed by second strand cDNA synthesis using DNA Polymerase I and RNase H treatment. The cDNA fragments then went through an end repair process, the addition of a single 'A' base, and then ligation of the Illumina multiplexing adapters. The products were then purified and enriched with PCR to create the final cDNA sequencing library. The cDNA library then undergoes quality control by running on the Agilent Bioanalyzer HS DNA assay to confirm the final library size and on the Agilent Mx3005P qPCR machine using the KAPA Illumina library quantification kit to determine concentration. A 2 nM stock was created and samples pooled by molarity for multiplexing. From the pool, 12 pmoles were loaded into each well for the flow cell on the Illumina cBot for cluster generation. The flow cell was then loaded onto the Illumina HiSeq 2500 utilizing v3 chemistry and HTA 1.8. The raw sequencing reads were processed through CASAVA-1.8.2 for FASTQ conversion and demultiplexing. The Illumina chastity filter was used and only the PF (passfilter) reads are retained for further analysis. Assembly of transcriptomes was performed using both Velvet/Oases and Trinity software packages with default settings (see list of commands subsection below).

List of commands used in sequence search:
Velvet/Oases.
velveth $outDir $hash_length -fastq -shortPaired $in_shuffled_ sequence_file
velvetg $outDir -read_trkg yes
oases $outDir -ins_length 150
Trinity
Trinity.pl –output $outDir –seqType fq –JM 90G –left $file1 –right $file2 –CPU 16

### Transmission electron microscopy

Animals were initially fixed whole in cold 2.5% glutaraldehyde in 0.1M cacodylate buffer, pH7.4 overnight in the refrigerator. After this initial fixation, the samples were stable enough so that small portions of selected areas could be dissected out and fixed for a further 24 hr at 4°C in 2.5% glutaraldehyde in 0.1M cacodylate. Following fixation, the samples were washed in 0.1M cacodylate buffer, incubated 1 hr in 1% osmium tetroxide at RT then washed with 0.1M cacodylate buffer, dehydrated through a graded ethanol series and embedded in epoxy resin. Semi-thin sections (0.5 microns) were cut, stained with toluidine blue and viewed by light microscopy to choose appropriate areas for study. Thin sections (70–80 nm) were cut from these selected areas and contrasted using 2% uranyl acetate and Reynold's lead citrate, and imaged on an FEI Tecnai T12 electron microscope.

## Isolation, purification, and analysis of collagen IV NC1 hexamers

Whole ctenophore tissues were frozen in liquid nitrogen, pulverized in a mortar and pestle and then homogenized in 2.0 ml g$^{-1}$ digestion buffer and 0.1 mg ml$^{-1}$ Worthington Biochemical bacterial collagenase and allowed to digest at 37°C, with spinning for 24 hr. Liquid chromatography purification of solubilized NC1 varied by species based on protein yield. All ctenophore NC1s were purified by gel-exclusion chromatography (GE Superdex 200 10/300 GL). For reduction and alkylation of collagen IV NC1 hexamers, fractions containing high-molecular-weight complex from size-exclusion chromatography were concentrated by ultrafiltration and reduced in TBS buffer with various concentrations of DTT. After incubation for 30 min at 37°C, samples were alkylated with twofold molar excess of iodoacetamide for 30 min at room temperature in the dark. After mixing with SDS loading buffer, samples were heated for 5 min in boiling water bath and analyzed by non-reducing SDS-PAGE. Collagenase-solubilized NC1 hexamers were analyzed by SDS-PAGE in 12% *bis*-acrylamide mini-cells with Tris-Glycine-SDS running buffer. Sample buffer was 62.5 mM Tris-HCl, pH 6.8, 2% SDS (w/v), 25% glycerol (w/v), 0.01% bromophenol blue (w/v). Western blotting of SDS-dissociated NC1 hexamer was developed with JK-2, rat monoclonal antibody (kindly provided by Dr. Yoshikazu Sado, Shigei Medical Research Institute, Okayama, Japan). All Western blotting in *Figure 6* was done with Thermo-Scientific SuperSignal West Femto chemiluminescent substrate and digitally imaged with a Bio-Rad GelDoc.

## Immunohistochemistry of ctenophore tissues

Whole ctenophore tissues were placed in 150 mL beaker and as much liquid was removed as possible. Each tube with tissue was filled with 100 mL ice-cold ctenophore fixation buffer 1 [80ul Glutaraldehyde (25%), 0.02% final concentration; 25 mL Paraformaldehyde (16%), 4.0% final concentration; 75 mL 0.2um-filtered seawater (Red Sea Coral Pro Salt)], inverted a few times gently, and left at 4 degrees Celsius for 5 min. Buffer 1 was then removed, and 100 mL of ctenophore fixation buffer 2 was added [25 mL Paraformaldehyde (16%), 4.0% final concentration; 75 mL 0.2um-filtered seawater (Red Sea Coral Pro Salt)]. Buffer 2 was then removed and tissues were gently washed five times with cold 1X PBS. Fixation protocol adopted from Pang and Martindale, *Ctenophore Whole-Mount Antibody Staining* (*Pang and Martindale, 2008*). Tissues were then embedded in parafilm and sectioned onto individual slides for IHC staining. After deparaffinization and rehydration, tissues underwent heat-induced epitope retrieval with DAKO and microwaved for 15 min. Cold tap water was then run over tissues for 10 min, followed by two washed with 1X PBS and stored in 1X PBS. Immunostaining occurred at room temperature, with blocking by a 5% serum blocking buffer (1X PBS pH 7.4/5% normal goat serum/0.1% Triton X-100) for 60 min. All IHC for collagen IV was conducted with the rat monoclonal antibody (mAb) JK-2, and antibody dilution was done in 5% serum blocking buffer accordingly. Alexa488 tagged anti-rat secondary was used for the fluorochrome-conjugated secondary antibody (RRID:AB_10893331), and dilution was also done in 5% serum blocking buffer. IHC images were taken on a Zeiss Axioplan microscope. Lenses used were a 20X lens (Plan-APOCHROMAT 20X/0,75; ∞/0,17) and a 40X lens (Plan-NEOFLUAR 40X/0,75; ∞/0,17). Images were taken at room temperature, approximately 20 degrees Celsius. All images were done in an imaging medium of air. Fluorochromes used were Alexa488 (green) for collagen IV, and Hoescht stain (blue) was used for nuclei staining. The Camera for imaging was a Photometrics CoolSnap HQ, using Metamorph 7.7.0.0 software (RRID:SCR_002368). Slight gamma correction of (< ± 0.2) after acquisition to adjust contrast. Images captured were merged with ImageJ64, 1.48v (RRID:SCR_003070).

## Phylogenetic analysis of NC1 domains

The evolutionary relationship between collagen IV and spongins was analyzed using the NC1 domains of each of the 139 sequences in our dataset. NC1 domains were aligned using the Geneious alignment tool within Geneious software package (RRID:SCR_010519), version 8.1.9 with default settings (*Silvestro and Michalak, 2012*). The resulting sequence alignment, which was 881 amino acid sites in length, was used to reconstruct the phylogeny of NC1 domains under the maximum likelihood optimality criterion as implemented in the RAxML software (RRID:SCR_006086), version 8.2.3 (*Stamatakis, 2014*). The phylogenetic analysis was performed using the PROTGAMMAAUTO option, which selects the substitution model with the best fit to the alignment among a set of among a set of 11 models (these were: DAYHOFF, DCMUT, JTT, MTREV, WAG, RTREV, CPREV, VT,

BLOSUM62, MTMAM, and LG). In the case of the NC1 domain phylogeny, the model with the best fit was the VT model (*Müller and Vingron, 2000*). Robustness in phylogeny inference was assessed with 100 bootstrap replicates.

## Acknowledgements

The technical work of Neonila Danylevych is greatly appreciated. We thank Carl Luer and the Mote Marine Laboratory for assistance with *Mnemiopsis* field collections. We acknowledge the Vanderbilt Technologies for Advanced Genomics (VANTAGE) for technical work in transcriptome assemblies. Electron microscopy was carried out in part through the use of the VUMC Cell Imaging Shared Resource (supported by NIH grants CA68485, DK20593, DK58404, DK59637 and EY08126). We would like to thank Dr. Yoshikazu Sado (Shigei Medical Research Institute, Okayama, Japan) for kindly providing the collagen IV JK-2 monoclonal antibody. This work counts in part towards the doctoral dissertation of ALF. at Tennessee State University. The authors declare no competing financial interests.

## Additional information

### Competing interests

AR: Reviewing editor, *eLife*. The other authors declare that no competing interests exist.

### Funding

| Funder | Grant reference number | Author |
| --- | --- | --- |
| National Institutes of Health | DK18381 | Billy G Hudson |
| National Science Foundation | DEB-1442113 | Antonis Rokas |
| March of Dimes Foundation | March of Dimes Prematurity Research Center Ohio Collaborative | Antonis Rokas |
| Aspirnaut Program | | Julie K Hudson Billy G Hudson |

The funders had no role in study design, data collection and interpretation, or the decision to submit the work for publication.

### Author contributions

ALF, CED, Conceptualization, Formal analysis, Investigation, Visualization, Writing—original draft, Writing—review and editing; SVC, Conceptualization, Resources, Data curation, Formal analysis, Investigation, Writing—original draft, Writing—review and editing; VKP, Formal analysis, Validation, Investigation, Methodology, Writing—review and editing; SPB, KLB, Resources, Formal analysis, Writing—review and editing; WGJ, Resources, Formal analysis, Methodology, Writing—review and editing; JKH, Conceptualization, Resources, Funding acquisition, Writing—review and editing; AR, Conceptualization, Resources, Funding acquisition, Methodology, Writing—review and editing; BGH, Conceptualization, Supervision, Funding acquisition, Visualization, Writing—original draft, Project administration, Writing—review and editing

### Author ORCIDs

Aaron L Fidler, http://orcid.org/0000-0002-2519-8864
Antonis Rokas, http://orcid.org/0000-0002-7248-6551
Billy G Hudson, http://orcid.org/0000-0002-5420-4100

## Additional files

### Supplementary files

• Supplementary file 1. Intron/exon boundaries and split-glycine codons in *Mnemiopsis*. (A) The human collagen IV one gene consists of 52 exons. The number of exons in the *Mnemiopsis* collagen

IV genes ranges from 6 to33. A characteristic feature of collagen IV is the presence of split glycine codons. Eleven of the collagen IV genes possess split glycine codons. (B) Intronic regions are in low-ercase and exon regions are in bolded UPPERCASE, underlined nucleotides represent partial codons, Green 'G' in the residue column denote glycines encoded by a split codon.

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
