## [Decision Letter]

Thank you for submitting your article "Collagen IV and the evolutionary dawn of metazoan tissues" for consideration by *eLife*. Your manuscript has now been reviewed by two reviewers and a Senior Editor. The following individuals involved in review of your submission have agreed to reveal their identity: Kevin P Campbell (Reviewer #2).

While there was general enthusiasm about this study, there are a number of issues that require attention, with the need for submission of a fully responsive revised manuscript that will again be evaluated by the reviewers.

One shared opinion relates to the choice and limited number of organisms that were assessed for collagen IV. Were other eukaryotes analyzed? Why was only one choanoflagellate checked? There are two genomes available: *Monosiga brevicollis* and *Salpingoeca rosetta* that should be included. The list could extend to opisthokonts (such as *Capsaspora owczarzaki*), as well as Amoebozoa and Apusomonadida (*Thecamonas trahens*). Along similar lines, it is interesting that you did not see an organized basement membrane in *Mnemiopsis* even though collagen IV and laminin are expressed. One possible explanation for this result is that *Mnemiopsis* lacks a critical collagen IV or other basement membrane receptor. If there are any data on the expression of various basement membrane receptors in *Mnemiopsis, Beroe* and *Pleurobrachia* it would be good to include this information in the manuscript.

It would be useful to provide a table with the gene content related to ECM in each of the taxa analyzed. There was also a request for greater explanation of phylogenetic methods that were employed in the study. How were sequences aligned and how many amino acid positions were included at the end? The bootstrap is explained, but how did you select the best, final tree? Which model of evolution was used? Was the LG model of evolution assessed? The trees shown in the figures are without branch lengths without explanation?

There was also an opinion that you should consider alternative evolutionary scenarios for the origin of collagen IV and the possibility that sponges diverged prior to Ctenophora and Bilateria. What would that imply?

---

## [Author Response]

*While there was general enthusiasm about this study, there are a number of issues that require attention, with the need for submission of a fully responsive revised manuscript that will again be evaluated by the reviewers.*

*One shared opinion relates to the choice and limited number of organisms that were assessed for collagen IV. Were other eukaryotes analyzed? Why was only one choanoflagellate checked? There are two genomes available: Monosiga brevicollis and Salpingoeca rosetta that should be included. The list could extend to opisthokonts (such as Capsaspora owczarzaki), as well as Amoebozoa and Apusomonadida (Thecamonas trahens).*

We have now extended unicellular taxa sampling beyond *Monosiga brevicollis* only, to include *Salpingoeca rosetta* (Choanozoa), *Capsaspora owczarzaki* (Filasterea), *Dictyostelium discoideum* (Amoebozoa), and *Thecamonas trahens* (Apusozoa). Our analysis indicates that collagen IV is absent in these species, as in *Monosiga brevicollis*. Further analysis of *S. rosetta* and *D. discoideum* revealed the presence of Gly-X-Y repeats, as in the case of *Monosiga*. In the revision, we present this information with a new ECM gene content figure (Figure 2) in the main text.

*Along similar lines, it is interesting that you did not see an organized basement membrane in Mnemiopsis even though collagen IV and laminin are expressed. One possible explanation for this result is that Mnemiopsis lacks a critical collagen IV or other basement membrane receptor. If there are any data on the expression of various basement membrane receptors in Mnemiopsis, Beroe and Pleurobrachia it would be good to include this information in the manuscript.*

In response, we analyzed collagen IV receptors in Ctenophora, along with other metazoans and protists, by identifying the presence or absence of integrins, dystrophin, and Discoidin domain receptors 1 and 2 (DDR). This information is presented in the new Figure 2 (ECM gene content figure). Integrins were found throughout all metazoans analyzed, including sponges and ctenophores, and are also present in the unicellular eukaryotes, *Capsaspora* and *Thecamonas* (Sebe-Pedros et al. Proc Natl Acad Sci U S A; 107: 22, 10142-7, DOI: 10.1073/pnas.1002257107). Dystrophins, while present in the unicellular eukaryotes, are absent in the ctenophores, *Mnemiopsis, Beroe,* and *Pleurobrachia,* as well as *Oscarella* sponge, and *Trichoplax*. DDR1 and 2 was found to be absent in all non-bilaterian species. Importantly, the results of collagen receptors in non-BM metazoans versus metazoans with BM are inconclusive in determining why *Mnemiopsis* or even *Trichoplax*, while containing collagen IV and laminin,does not form basement membrane. Likely, there is a yet unidentified component that plays a role in assembly of basement membranes.

*It would be useful to provide a table with the gene content related to ECM in each of the taxa analyzed.*

We have included an ECM gene content table as a new main figure, Figure 2, which summarizes the ECM gene content across each of the taxa analyzed and compliments the ECM component data presented in Figure 1.

*There was also a request for greater explanation of phylogenetic methods that were employed in the study. How were sequences aligned and how many amino acid positions were included at the end? The bootstrap is explained, but how did you select the best, final tree? Which model of evolution was used? Was the LG model of evolution assessed? The trees shown in the figures are without branch lengths without explanation?*

Our initial tree was constructed using BLOSUM62 matrix (original Figure 8—figure supplement 1). We conducted additional analyses using the LG model along with ten other models of evolution (DAYHOFF, DCMUT, JTT, MTREV, WAG, RTREV, CPREV, VT, BLOSUM62, MTMAM). The topology of the tree depicting NC1 domain phylogeny remained unchanged, however the VT model yielded the tree with the best-fit. We replaced the BLOSUM62 tree with the VT model tree (new Figure 8—figure supplement 1).

We have revised the manuscript text as follows:

“We conducted additional phylogenetic analysis of the NC1 domain using RAxML to select the best tree from eleven models of evolution (DAYHOFF, DCMUT, JTT, MTREV, WAG, RTREV, CPREV, VT, BLOSUM62, MTMAM, and LG). Among these, the VT model yielded the tree with the best-fit (Figure 8—figure supplement 1).”

Additionally, we revised the Methods section, under “Phylogenetic analysis of NC1 domains”, to read:

“The evolutionary relationship between collagen IV and spongins was analyzed using the NC1 domains of each of the 139 sequences in our dataset. […] Robustness in phylogeny inference was assessed with 100 bootstrap replicates.”

*There was also an opinion that you should consider alternative evolutionary scenarios for the origin of collagen IV and the possibility that sponges diverged prior to Ctenophora and Bilateria. What would that imply?*

We revised the Discussion to include the possibility of sponges diverging prior to Ctenophora. Our proposed collagen IV evolutionary model is compatible with either hypothesis, Ctenophora or Porifera-first. The main difference being the order in which the early metazoan ancestor collagen IV gene duplicated. Namely, it either, a), first duplicated to two chains in the Porifera lineage and underwent several subsequent gene duplication events later in Ctenophora, or, b), the early metazoan ancestor collagen IV chain underwent several gene duplication events in the Ctenophora lineage, and then through genetic streamlining, these multiple chains condensed to only the two chains that are found in the other non-bilaterian metazoan phyla. We presented these possibilities in the text and with a new figure, Figure 10, as well as a revision to the collagen IV evolution model figure, Figure 11 (the original Figure 9).